# A Functional End-Use of Avocado (cv. Hass) Waste through Traditional Semolina Sourdough Bread Production

**DOI:** 10.3390/foods12203743

**Published:** 2023-10-11

**Authors:** Enrico Viola, Carla Buzzanca, Ilenia Tinebra, Luca Settanni, Vittorio Farina, Raimondo Gaglio, Vita Di Stefano

**Affiliations:** 1Department of Agricultural, Food and Forest Sciences (SAAF), Università degli Studi di Palermo, Viale delle Scienze, 90128 Palermo, Italy; enrico.viola01@unipa.it (E.V.); luca.settanni@unipa.it (L.S.); vittorio.farina@unipa.it (V.F.); raimondo.gaglio@unipa.it (R.G.); 2Department of Biological, Chemical and Pharmaceutical Science and Technology (STEBICEF), University of Palermo, Via Archirafi, 90123 Palermo, Italy; carla.buzzanca@unipa.it (C.B.); vita.distefano@unipa.it (V.D.S.); 3Centre for Sustainability and Ecological Transition, University of Palermo, Piazza Marina, 90133 Palermo, Italy

**Keywords:** avocado wastes/by-products, functional bread, lactic acid bacteria, sourdough, peels, pulp, seeds, polyphenols, antioxidant properties

## Abstract

In recent years, a main goal of research has been to exploit waste from agribusiness industries as new sources of bioactive components, with a view to establishing a circular economy. Non-compliant avocado fruits, as well as avocado seeds and peels, are examples of promising raw materials due to their high nutritional yield and antioxidant profiles. This study aimed to recycle avocado food waste and by-products through dehydration to produce functional bread. For this purpose, dehydrated avocado was reduced to powder form, and bread was prepared with different percentages of the powder (5% and 10%) and compared with a control bread prepared with only semolina. The avocado pulp and by-products did not alter organoleptically after dehydration, and the milling did not affect the products’ color and retained the avocado aroma. The firmness of the breads enriched with avocado powder increased due to the additional fat from the avocado, and alveolation decreased. The total phenolic content of the fortified breads was in the range of 2.408–2.656 mg GAE/g, and the antiradical activity was in the range of 35.75–38.235 mmol TEAC/100 g (*p* < 0.0001), depending on the percentage of fortification.

## 1. Introduction

Avocado (*Persea americana* Mill.) is a subtropical/tropical fruit native to Mexico and Central America and is widely produced and consumed worldwide [1]. In recent years, avocado production has steadily increased globally due to the growing popularity and demand for the fruit [2]. The main avocado producers are Mexico (33%), the Dominican Republic (10.5%), Peru (7.8%), Indonesia (5.7%), and Colombia (5.1%) [3]. Spain and Italy are the only European countries with significant commercial production of avocados, which are cultivated, respectively, on the Andalusian Mediterranean coast, mainly in the provinces of Malaga and Granada, and in Sicily, along the Tyrrhenian coastal areas and close to Catania [4]. In Europe, avocado consumption per capita increased by an average of 180% between 2012/13 and 2018/20, with industry expectations for further increases [5]. A significant portion of this demand is driven by the young millennial generation in Europe and increased consumer interest in so-called “superfoods”. Avocado fruit has great potential to meet consumers’ desired requirements due to its high nutritional value, particularly its antioxidants, fiber, and low sugar content [6,7,8]. For these reasons, eating avocados is generally recommended for people with diabetes because it is a high-energy food [9] and can be used in a wide range of food products [10]. However, consumers’ avocado preference may depend on several quality attributes [11,12]. These quality attributes refer to physical product characteristics such as freshness, color, and size and experience attributes such as taste, aroma, and stage of maturation [13,14]. External factors that most influence consumer choice include fruit weight (commercial size), peel color (green or black), absence of defects (crick side, blanch, terminal spot), and ripening stage, which is closely related to fruit firmness, given the high perishability of avocados [14,15]. For these reasons, consumers prefer to buy unripe and/or not fully ripe avocado fruit [14], avoiding fruit that is already ripe or overripe.

Avocados are mostly consumed fresh, but they are also processed to extract oil and other products, such as guacamole [16]. Therefore, several components of the fruit, including the peel and seeds, are not used and are wasted, becoming a source of environmental contamination. However, these components are rich in protein, fiber, and numerous bioactive compounds [17,18]. For instance, the seed and peel of the “Hass” avocado account for about 15% and 14% of the weight of the fruit, respectively [18,19,20]. This is equivalent to at least 1.6 million tons of avocado seeds and peels annually discarded worldwide [1], which adds to the global share of food waste. Among the various processing techniques that can be used to recycle both the discarded (due to overripeness) avocado fruit and its by-products is dehydration.

Drying is probably one of the oldest methods of food preservation [21] and consists of the removal of water to a final concentration, which assures microbial stability and ensures the expected shelf-life of the product [22]. In addition, this technique is the most widely used for creating powders from fresh fruits [23].

Fruit and vegetable powders can be used as intermediates in the beverage industry, functional food additives that improve the nutritional value of foods, flavoring agents, or natural coloring agents [24]. Fruit and vegetable powders also serve as ingredients for pasta, breads, dry soups, and other food recipes [25,26,27,28,29].

Powder quality depends largely on the drying and milling conditions as well as the composition and quality of the raw material [30,31].

Fruit and vegetable powder ingredients for dough and/or bread preparation must be strategically selected to achieve the optimal composition and physical properties and avoid adverse effects [32]. In fact, vegetable powder can potentially decrease the stability of dough because the fiber in it slows down the rate of hydration and gluten development. This depends on the amount of vegetable powder incorporated [33]. Similarly, vegetable powder may affect texture differently depending on the type of by-product. On the other hand, fruit and vegetable powders can impart coloring and stabilizing properties to the final product due to the presence of carotenoids and polyphenols [33]. This work aimed to recycle avocado waste to produce new value-added ingredients. To this end, avocado waste was dehydrated and milled. The resulting powder was then used as ingredients for processing sourdough semolina bread to functionalize this food, widely consumed daily in Southern Italy.

## 2. Materials and Methods

### 2.1. Chemicals and Reagents

Methanol, sodium carbonate, gallic acid, Folin-Ciocalteu′s phenol reagent, DPPH (2,2-diphenyl-1-picrylhydrazyl), ABTS (2,2′ azino-bis (3-ethylbenzothiazoline-6-sulfonic-acid), potassium persulphate, sodium hydroxide (NaOH) and Trolox (6-hydroxy-2,5,7,8-tetramethylchroman-2-carboxylic acid) were obtained from Fluka (Buchs, Switzerland). HPLC-grade water was obtained by purifying double distilled water in a Milli-Q Gradient A10 system (Millipore, Bedford, MA, USA), 0.45 µm PTFE syringe filter (Whatman, Milan, Italy).

### 2.2. Production of Avocado Waste Powder (AWP) and Commercial Semolina

Avocado fruit (*Persea americana* Mill.) cv. Hass was harvested at the experimental field of the Department of Agricultural, Food, and Forestry Sciences, University of Palermo. After being harvested, the fruits were left to ripen at a temperature of 20 ± 5 °C, and the progress of ripening was assessed by the change in epicarp color; therefore, the shade angle indicator was used, as described by Sánchez-Quezada et al. [34].

To obtain a uniform sample representative of the ripening stage, the researchers chose to pick avocado fruit at the overripe stage, i.e., fruits with hue angle values of ≥45 ± 7 h°, as determined by Sánchez-Quezada et al. [34]. After being sanitized in chlorinated water (2% *v*/*w*) for 10 min, the avocado fruits were peeled and their seeds were removed.

The different components of the avocado fruit (pulp, seeds, and peels) were separated as they required different dehydration times and temperatures. After several preliminary tests the avocado pulp and its by-products were dried as follows:-Pulp at a temperature of 75 °C for 28 h;-Peel and seeds at a temperature of 60 °C for 4 h.

Before dehydration, the seeds were washed with water, and the outer covering of the seeds was removed manually during washing. A tray dryer (Ausla, 1000 Watt, Milan, Italy) was used to dry the pulp and by-products.

The time/temperature binomials chosen generated a moisture content of less than 12–13%, which is the range selected to avoid microbial proliferation and to achieve rapid drying that would not lead to degradation of the bioactive components [35]. The dehydrated products obtained (pulp, seed, and peel) were separately processed into “powder” by an ultra-centrifugal mill (Fritsch, Pulverisette 14, Lainate, Italy). To obtain a powder with particle sizes between 1.5 and 2 mm from each fruit part (pulp, seeds, and peels), they were processed at 700 rpm for 10 s.

For breadmaking, a “powder mixture” (Avocado Waste Powder: AWP) comprised of pulp, seeds, and peels was used. The AWP consisted of the following percentages of dried fruit: 50% pulp, 25% seeds, and 25% peel.

A commercial semolina (Cuore Mediterraneo, Santa Giusta, Italy) was used to process the bread for this study. Its nutritional values (per 100 g) were: 12.5 g of protein; 1.5 g of fat; 0.3 g of saturated fats; 69 g of carbohydrates, and 26 g of fiber.

### 2.3. Determination of Color Characteristics of AWP

The color of both semolina and AWP samples was measured using a Minolta colorimeter (Chroma Meter CR-400, Konica Minolta Sensing Inc., Tokyo, Japan), and the L* (brightness), Chroma (C*), and hue angle (h°) parameters were evaluated [21]. The instrument was calibrated using a standard white plate. Chroma (C*) values and hue angles (h°) were calculated using Equations (1) and (2), respectively:(1)$$C^{*}={{(a}^{2}+b^{2})}^{\frac{1}{2}}$$
(2)$$h{}^{\circ}=arctang\left(\frac{b}{a}\right)$$

Using the obtained values of L*, a*, and b*, a color table was created by converting the CIEL*a*b* color space to the red/green/blue (RGB) scale through the e-paint.co website (accessed on 15 June 2023).

### 2.4. Hygienic Characteristics of AWP

The AWP was microbiologically analyzed for some microbial groups that are unwanted during food fermentation, as reported by Messina et al. [36]. Briefly, 10 g of AWP was first homogenized by a BagMixer^®^ 400 stomacher (Interscience, Saint Nom, France) and then serially diluted. The diluted samples were analyzed for the following microbial groups: total mesophilic microorganisms (TMM), members of the Enterobacteriaceae family, total coliforms, and spore-forming aerobic bacteria. The analyses were performed in duplicate.

### 2.5. Bacterial Strains

Lactic acid bacteria (LAB) isolated from Sicilian sourdoughs and previously tested to produce semolina breads with the addition of by-product ingredients [37] were used to prepare a multiple-strain sourdough starter. The strains *Lentilactobacillus diolivorans* SD4, *Fructilactobacillus sanfranciscensis* SD22, *Levilactobacillus brevis* SD46, *Lactiplantibacillus plantarum* SD96, *Weissella cibaria* SD123, *Lactiplantibacillus pentosus* SD130, *Leuconostoc citreum* SD142, and *Leuconostoc holzapfelii* SD148, all belonging to the Culture Collection of the Agricultural Laboratory of the University of Palermo, Italy, were defrosted from −80 °C and cultivated in de Man-Rogosa-Sharpe medium modified as described by Lhomme et al. [38] at 30 °C for 24 h.

### 2.6. Sourdough Propagation

After reactivation in a synthetic medium, all the LAB strains were propagated in sterile semolina extract (SSE) broth [39]. Commercial semolina was used to both prepare liquid SSE broth and propagate solid sourdough. The individual cultivation of LAB and the mixed cell culture representing sourdough inoculum were performed as reported by Gaglio et al. [40].

The LAB mixed culture was diluted in sterile tap water to reach a final volume of 187.5 mL. This cell suspension was added to 312.5 g of semolina to obtain a 500 g dough with a dough yield (DY = weight of the dough/weight of semolina × 100) of 160 and a cell density of about 10^6^–10^7^ CFU/g. The dough was then left to ferment at 28 °C for 16 h and subjected to seven consecutive daily refreshments to generate a mature sourdough inoculum [41].

### 2.7. Bread Doughs and Baking Process

Bread production was carried out solely with the sourdough developed from the selected LAB strains. No baker’s yeast and kitchen salt were added to evaluate the effect of AWP on the performance of LAB. The control (CTR) doughs (800 g) to be leavened before baking were processed by adding 228.6 mL of sterile tap water and 457.2 g of semolina to 114.2 g of mature sourdough (DY = 175). The experimental AWP doughs were produced with the same amount of water and sourdough, but the amount of semolina was reduced to 434.3 g and 411.4 g for the 5-AWP [containing 5% (*w*/*w*) AWP] and 10-AWP [containing 10% (*w*/*w*) AWP] trials, respectively. A planetary mixer (model XBM10S; Electrolux Professional SpA, Pordenone, Italy) was used to mix all the ingredients for 15 min with a paddle turning on Speed 4. Aliquots of 100 g per dough were transferred into trapezoidal stainless steel baking pans [42], kept at 28 °C for 8 h, and then baked as reported by Alfonzo et al. [43]. Two technical repeats were obtained from each bread trial (performed in duplicate), and all bread baking was repeated after two weeks to obtain two independent replicates.

### 2.8. Acidification Process

Sourdough fermentation was monitored by pH measurement, total titratable acidity (TTA) determination, and the evolution of LAB numbers following the approach of Francesca et al. [44]. To perform LAB viable counts and that of other microbial groups relevant during dough fermentation, the sourdough and bread doughs were microbiologically evaluated to enumerate TMM, sourdough LAB, yeasts, members of the Enterobacteriaceae family, and total coliforms, as reported by Gaglio et al. [37]. All analyses were performed in duplicates.

### 2.9. Quality Characteristics of Breads

The breads were cooled at room temperature for approximately 30 min after baking and investigated for several quality parameters, as reported by Cirlincione et al. [45]. In particular, the following parameters were considered: weight loss (WL, %), specific volume (cm^3^/g bread), firmness (N/mm^2^), crust and crumb color [Lightness (L*), redness (a*) and yellowness (b*)], void fraction (%), cell density (number of cells/cm^2^), and mean cell area (mm^2^). The analyses were performed in duplicate.

### 2.10. Chemical Characterization

#### 2.10.1. Total Phenolic Content Analysis

Total phenolic content (TPC) was determined using the optimized Folin–Ciocalteu method previously published [46]. One gram of each bread sample (CTR-Bread, bread produced with control dough; 5-AWP Bread, experimental bread enriched with 5% (*w*/*w*) of avocado waste powder (AWP); 10-AWP Bread, experimental bread enriched with 10% (*w*/*w*) of AWP) was added to 5 mL of methanol/water (80:20 *v*/*v*) and sonicated and filtered through Whatman 0.45 μm PTFE filters. This was followed by a reaction with the Folin-Ciocalteu reagent in the presence of sodium carbonate to form a blue-colored complex. The intensity of the color was proportional to the phenolic compounds in the sample. The resulting colorimetric reaction was measured at 765 nm using a UV-VIS spectrophotometer (Varian Cary 50, Agilent, Santa Clara, United States). The amount of TPC was calculated by interpolation from a calibration curve of gallic acid [0.001 to 0.25 mg/mL] (y = 10.945x + 0.1305, R^2^ = 0.993). The results were expressed as mg gallic acid equivalents per g (mg GAE g^−1^) of the sample.

#### 2.10.2. Radical Scavenging Properties Evaluation, DPPH and ABTS Assay

The measurement of the powder and fortified bread samples’ antiradical activity (DPPH (2,2-diphenyl-1-picrylhydrazyl) and ABTS (2,2′ azino-bis (3-ethylbenzothiazoline-6-sulfonic-acid) assays) followed a procedure previously described by Di Stefano et al. [47]. The DPPH assay was used for the in vitro evaluation of the scavenger activity toward free radicals. One g of each bread sample (AWP and semolina) was extracted with 4 mL of methanol, mixed with 3 mL of DPPH (60 μM) and placed in the dark for 30 min. Scavenging activity was monitored by spectrophotometric analysis of the absorbance at a wavelength of 517 nm with a UV-VIS spectrophotometer (Varian Cary^®^ 50, Agilent, Santa Clara, United States) and using methanol as the blank. The results were reported as Trolox equivalent antioxidant activity and expressed as mmol Trolox equivalent (TE)/100 g of the sample. The absorbance signal was translated into antioxidant activity using Trolox as the standard and the calibration curve in the range of 5–400 μM (y = −0.0008x + 0.4036, R^2^ = 0.998). All experiments were performed in triplicate. According to Re et al. [43], for the ABTS assay, the ABTS^+^ radical cation was produced by reacting ABTS stock solution with 2.45 mM potassium persulfate; the solution turned a dark blue-green at the end of the reaction time. One gram of each sample was added to 4 mL of methanol, sonicated, and filtered through Whatman 0.45 μm PTFE filters. A calibration curve using Trolox at increasing concentrations [2.5–30 µM] was constructed. The assays were performed in triplicate.

### 2.11. Sensory Analysis

A panel of 17 judges, including 10 women and seven men whose ages ranged from 26 to 57, was recruited to perform a descriptive sensory analysis of the breads produced with different percentages of AWP. First, the judges were trained to acquire familiarity with bread attributes by tasting commercial semolina bread. The judges were asked to score appearance, texture, odor, and taste descriptors from among those reported by Ruisi et al. (2021) [48]. The evaluation for each attribute was expressed on a 9-point scale (1 = extremely bad; 9 = extremely good). The evaluation was performed by the judges in individual chambers following the ISO 13299 guidelines [49].

### 2.12. Statistical Analysis

Differences between the microbiological and physicochemical data were identified by way of one-way variance analysis, while Tukey’s test was used for multiple mean comparisons (statistical significance *p* < 0.05). In addition, a hierarchical cluster analysis (HCA) was performed to group the produced breads according to their dissimilarity, as reported by Martorana et al. [50]. The data were statistically processed using XLStat software version 7.5.2 for Excel (Addinsoft, New York, NY, USA).

## 3. Results

### 3.1. Color Characteristics

From the analysis of colorimetric data (presented in Table 1), the semolina sample showed an elevated brightness (86.90 ± 1.63) compared to the AWP sample but a lower C* index. This was due to the color of the semolina, which, as can be seen from the RGB data, tended toward white. The AWP, on the other hand, showed lower values of L* but higher values of Chroma. From the values obtained and the evaluation of the color table, the powders maintained a color tending toward green. Therefore, processing the dehydrated avocados into powder did not alter their color.

### 3.2. Monitoring of the Fermentation Process

The growth of the selected LAB strains in SSE showed a consistent decrease in pH until the average value of 4.03 ± 0.10. The lowest pH (3.82 ± 0.18) was reached by the strains *L. pentosus* SD130 and *L. plantarum* SD96. The mixture with the eight LABs developed individually in SSE that was obtained after three propagation days was used as a liquid inoculum to produce sourdough for bread production. Soon after inoculation of semolina and tap water with the eight strains, the obtained dough showed a pH of 5.44 ± 0.21; this value decreased to 3.92 ± 0.04 in the sourdough obtained after seven days with daily refreshments. The TTA of the mature sourdough was 12.10 ± 0.07 NaOH 0.1 N/10 g. Table 2 presents the pH and TTA data for the unbaked doughs up to 8 h of leavening. A pH value of 5.5 was measured for the CTR dough at the start of fermentation.

The initial pH of the 5-AWP dough was slightly lower (5.42) than that of the 10-AWP dough (5.56) and the CTR dough. During fermentation, the pH values decreased progressively until they reached almost similar values of 4.19, 4.33, and 4.37 for the CTR, 5-AWP, and 10-AWP doughs, respectively, at the end of the monitoring period. An inverse trend was noted for the TTA, which increased linearly over time. After 8 h of fermentation, the TTA of the CTR dough (8.93 mL NaOH 0.1 N/10 g) was slightly higher than the value expressed by the AWP doughs (8.65 and 8.60 for 5-AWP and 10-AWP, respectively). The results of the plate count of the doughs are presented in Table 3. Sourdough developed from LAB selected at the seventh refreshment was characterized by 8.64 and 7.74 Log CFU/g of LAB and yeasts, respectively. Regarding the doughs leavened for bread production, LAB accounted for 7.60–7.71 Log CFU/g soon after ingredient mixing. These data were a little higher than those shown by the TMM (6.60–6.66 Log CFU/g) and confirmed that LAB from the sourdough inoculum was significantly transferred to the bread doughs. The levels of yeast immediately after production (0 h) were one order of magnitude lower than the LAB. After 8 h of fermentation, the cell densities of LAB increased in all the trials. The LAB levels of the CTR and 5-AWP doughs were almost comparable (8.94 and 8.98 Log CFU/g, respectively), while a slightly lower density (8.51 Log CFU/g) characterized the 10-AWP dough. A very limited increase in cell density was registered for the yeasts, barely overcoming 7.0 Log CFU/g for the CTR and 5-AWP doughs at the end of fermentation. Regarding hygiene indicators, although the levels of Enterobacteriaceae and total coliforms in the AWP were below the detection limit (for this reason, these results are not included in Table 3), after 8 h of fermentation, their presence was revealed in both AWP bread doughs at levels around 10^3^ CFU/g. No spore-forming bacteria were detected in the AWP and corresponding bread doughs.

### 3.3. Bread Quality Attributes

The characteristics of the final breads produced are summarized in Table 4.

Weight loss after baking was 11.30% in the CTR bread, while lower values were displayed by the AWP breads. The specific volume of the breads decreased with the different percentages of AWP; a value of 3.18 cm^3^/g was registered for the CTR bread compared to a value of 2.74 cm^3^/g for the 10-AWP bread. The addition of AWP determined a linear increase in firmness, with the highest value (0.113 N/mm^2^) recorded for the 10-AWP bread. Furthermore, the addition of AWP determined a change in the color parameters of both the crust and crumb of the breads, especially for L* and a*. Both parameters decreased progressively with the AWP percentages. Negative values were registered for the crumbs of all the trials. Image analysis of the breads indicated an increase in the void fraction and cell density of the crumbs with increasing percentages of AWP and a decrease in alveolation.

### 3.4. Chemical Characterization of Raw Materials and Bread Samples

The antioxidant activity and antiradical scavenging activity of the raw materials were measured; in particular, as shown in Table 5, high TPC was mostly highlighted in the AWP (197.775 mgGAE/g) compared to the semolina (3.676 mgGAE/100 g). The highest increase in antiradical activity was observed in the AWP, with values of 38.235 mmol TE/100 g and 35.175 mmol TE/100 g for the DPPH and ABTS assays, respectively, while the lowest was recorded for semolina (2.656 and 2.408 mmol TE/100 g for the DPPH and ABTS assays, respectively).

The same analyses were carried out on the bread samples fortified with different percentages of AWP. The addition of AWP enhanced the samples’ antiradical and antioxidant activity. As Table 6 shows, 10-AWP bread had higher values of antioxidant (23.882 mgGAE/100 g) and antiradical activity (6.656 and 9.234 mmol TE/100 g for DPPH and ABTS assays, respectively) compared to the CTR bread, which was made with only semolina.

### 3.5. Bread Sensory Attributes

Figure 1 presents the spider plot resulting from the sensory evaluations of the CTR and AWP breads.

The addition of AWP, at both percentages, greatly affected the sensory characteristics of the semolina breads. The sensory traits significantly different from those of the control breads were crust and crumb color; crispy crust; bread and strange odor; astringent, bitter taste persistency; bread and strange aroma; and, especially, aroma intensity. Except for bread odor and bread aroma, which were scored at a lower level than the CTR bread, all the other traits mentioned had higher scores for the AWP breads. Regarding bread structure, although alveolation and adhesiveness are lower in AWP breads, the differences are not significant. Considering the overall assessment based on all these traits, the CTR bread received the highest scores, and between the 5-AWP bread and 10-AWP bread trials, the breads processed with 5% AWP were more appreciated.

### 3.6. Multivariate Analysis

The HCA clustered the breads based on their dissimilarity and relationship using a total of 37 variables, including quality attributes, antioxidant and antiradical properties, and sensory traits. The resulting cluster presented in Figure 2 shows low levels of dissimilarity (0.099%) among the breads. However, the breads enriched with AWP formed a single cluster and were clearly separated from control production.

## 4. Discussion

In the production of cereal-based foods, fiber has grown in popularity as an added functional ingredient. Fiber, such as inulin, improves the rheological and technological characteristics of food, as well as its consistency, acceptability, and healthy properties. It also targets the prevention of metabolic syndromes. In addition to these improvements, when a food product is enriched with fiber, its shelf life is extended [51]. Other fiber-rich matrices, such as wheat or oat bran, have been used to replace wheat flour in baking [52]. One of the main reasons to supplement foods with dietary fiber is that it produces a wide variety of flavors that make products more palatable [53]. In a recent study, it was also shown that fortifying semolina bread with hemp seed flour improves its nutritional and antioxidant properties without significant changes in rheological properties [54]. In the work of Gómez and Martinez [32], the incorporation of fruit and vegetable by-products in baked products was evaluated to create foods with a higher fiber content. The authors highlighted a slowdown in the digestion of starch and other carbohydrates present in cereals and an improvement in rheological properties and interactions with digestive enzymes in the stomach. The improvement of antioxidant activity in fortified foods is due to an increase in the content of bioactive compounds, such as polyphenols and carotenoids [32]. Additionally, in the study of Gaglio et al. [40], the reuse of by-products for the production of fortified bread was evaluated. In particular, the authors used powdered almond skin at different percentages (5% and 10%) to produce functional products by modifying a traditional sourdough bread recipe. The final characteristics of the bread were influenced by the fortification and its percentage. The powdered almond skin positively influenced the sensory characteristics of the bread, with an increase in the intensity of the odor and the color of the crust and crumb. Thanks to the phytochemicals released by the fortified bread, an increase in the antioxidant capacity that can provide antioxidant protection at the level of human intestinal cells was also highlighted. Moreover, the microbiological parameters during fermentation were influenced by the development of coliforms due to the presence of spores after baking [37]. A recent study investigated Cava lees, another type of by-product that represents 25% of wine industry waste and is rich in antioxidant compounds and dietary fiber. This study aimed to evaluate the effect of Cava lees on microbial populations during natural leavening and bread fermentation. The results showed that the bread formulation with 5% Cava lees promoted the growth of both LAB and yeast and increased the concentration of volatile substances typically present [55].

Considering managing agro-wastes and food by-products while avoiding environmental concerns, the present work focused on valorizing the avocado production chain through the reuse of waste arising from non-compliant avocado fruits as well as avocado seeds and peels, to produce functional bread. From the results obtained in this study, it can be stated that the right time-temperature combination for the dehydration of avocado pulp and by-products, which maintained colorimetric and antioxidant characteristics after the dehydration process, was found. In particular, the grinding process did not affect or alter the organoleptic characteristics of the powder. It is important to emphasize that color is an extremely important characteristic because it makes the product attractive and acceptable, inducing consumers to purchase it; in other words, it is the first quality that guides consumers’ purchase choices [56]. The dehydration process tends to alter the surface characteristics of the food and, consequently, alters both reflectivity and color properties; particularly in fruit, alterations occur at the expense of carotenoids and chlorophyll. Such alterations were not observed in this study, and as reported in the results, the powders retained a greenish color reminiscent of fresh avocado. In addition, due to the dehydration process, loss of the aromatic substances could occur [57], depending on the amount of heat energy absorbed by the product, in the form of sensible and latent heat, for the vaporization of water. The amount of this loss depends on temperature, moisture content of the food, vapor pressure, and the solubility of the volatile compounds in water vapor. However, the extent of this loss is also related to lipid oxidation reactions [58]. The oxidation of fatty acids gives rise to aldehydes, ketones, and acids that cause rancidity and off-flavors [59]. In this case, the products obtained by the dehydration protocol used did not cause such alterations, and this is deducible from the sensory evaluation of the functional bread. For bread making, the sourdough inoculum was developed from selected starter strains. They all acidified the SSE used for pH values in the range of 3.82–4.34, which are generally registered for sourdough *Lactobacillus*, *Leuconostoc*, and *Weissella* grown in this semolina-derived medium [39,60]. Looking at the TTA data for the starter strains and bread doughs, this parameter was confirmed to evolve (increase) inversely with pH [61]. The acidification process was also followed through the LAB development, and at the end of fermentation (8 h), they increased about two Log cycles in all the bread doughs, as commonly observed in sourdough bread production [62]. The LAB levels were slightly higher than the TMM levels, indicating the absolute dominance of the added strains during fermentation. Furthermore, the low TMM levels registered are a consequence of the high nutritional requirements of LAB that are not fully satisfied by principal component analysis [63,64]. Of course, during sourdough fermentation, the development of yeasts is also particularly important [65]. Yeast cell densities estimated at the end of the leavening duration were between 6.78 and 7.06 Log CFU/g, although they were not deliberately inoculated. However, yeasts develop spontaneously in sourdough [66], and the results of this study are generally found in semolina sourdough fermentation [41], even in the presence of waste/by-product addition [40,64]. Furthermore, in this research, the ratio between yeasts and LAB was optimal at 1:100, which is considered optimal for sourdough preparations [67]. The AWP was also investigated for several undesired groups such as *Salmonella* spp., *Listeria monocytogenes*, coagulase-positive staphylococci, members of Enterobacteriaceae, coliforms, *Escherichia coli*, spore-forming bacteria, and *Pseudomonas*; none of these bacterial groups exceeded the detection limit. However, after 8 h of fermentation, members of Enterobacteriaceae and total coliforms, but no *E. coli*, were detected in both AWP bread doughs. Although undesirable in dough because they compete with LAB and yeast development, the levels estimated were particularly low, and this is imputable to the fact that Enterobacteriaceae are limited in their growth by the low pH encountered in sourdough during fermentation [68]. The quality attributes of the breads were impacted by the addition of AWP. In general, a diminution in WL and the specific volume of the breads is generally reported when food waste is added [37]. The data presented in this work confirmed this trend, but only the 10-AWP breads were characterized by a WL and specific volume significantly different from those displayed by the CTR bread. The browning of the crust and crumb was due to the darker color of AWP compared to semolina. This browning is a positive sign since, as reported by Sandvik et al. [69], the dark color of bread is linked to health among consumers.

The firmness of the breads increased as a consequence of the increase in dietary fiber, as observed by Ruisi et al. [48]. Regarding the image analysis of the central slices of the breads, alveolation diminished with the AWP-enriched breads. This phenomenon was also observed with the addition of pumpkin pomace and dry tomato waste [70,71] and is due to the low percentage of gluten [72]. The antiradical and antioxidant activity was mostly highlighted in the AWP breads compared to the semolina breads. The highest value of TPC was also observed in the AWP sample, while the lowest was recorded for the semolina sample. The supplementation of AWP in bread enhanced the food’s antiradical and antioxidant activity. The 10-AWP bread had higher values of antioxidant and antiradical scavenging activity compared to the CTR bread, which was made with only semolina. According to TPC values, the chemical analyses showed that the fortification of bread with AWP in different percentages increased its antiradical and antioxidant activity and organoleptic and baking qualities proportionally to the percentage of fortification used.

## 5. Conclusions

The overall data collected showed the excellent suitability of AWP for functional bread making. The present research found that avocado waste products that have been dehydrated and processed into powder can be successfully incorporated into leavened baked products to improve their characteristics, particularly in terms of antioxidant content. A positive relationship between the proportion of added powder and antioxidant content appeared, as well as the organoleptic and baking qualities of bread. The addition of 10% AWP produced dough with a higher antioxidant profile than the control bread. In addition, the bread produced from this dough was highly appreciated on a sensory level in terms of aroma and color. Fortified bread, therefore, has shown great potential to serve as a functional food among consumers.

## Figures and Tables

**Figure 1 foods-12-03743-f001:**
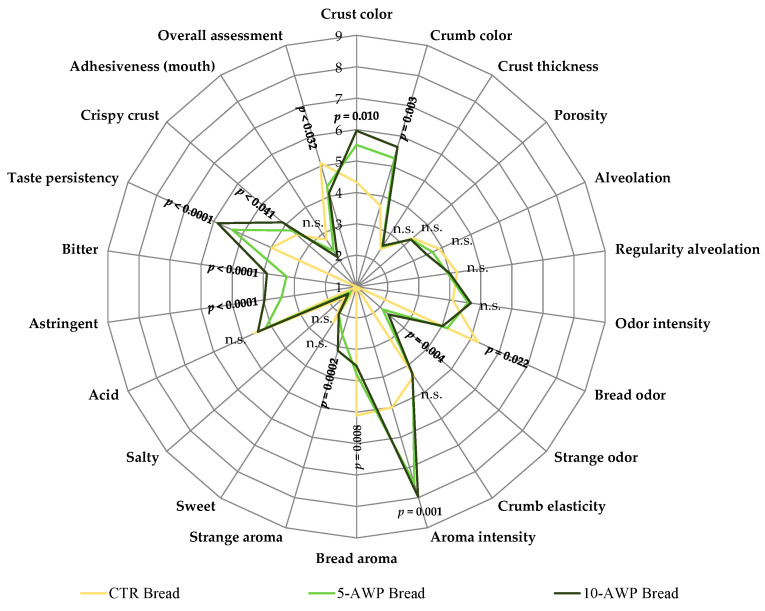
Spider diagrams of descriptive sensory analysis of breads. Abbreviations: CTR Bread; 5-AWP Bread, experimental bread enriched with 5% (*w*/*w*) of avocado waste powder (AWP); 10-AWP Bread, experimental bread enriched with 10% (*w*/*w*) of AWP; n.s., not significant (*p* > 0.05).

**Figure 2 foods-12-03743-f002:**
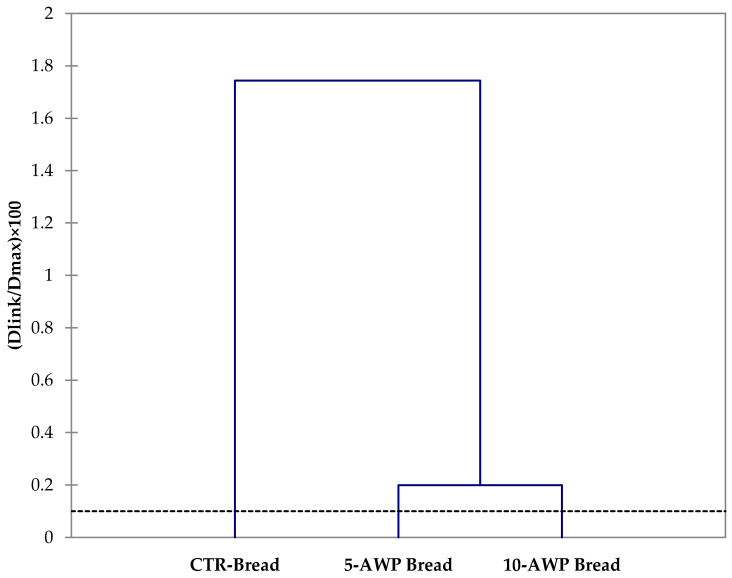
Dendrograms obtained from hierarchical cluster analysis based on values of quality attributes, antioxidant and antiradical properties and sensory traits of breads. Abbreviations: CTR Bread; 5-AWP Bread, experimental bread enriched with 5% (*w*/*w*) of avocado waste powder (AWP); 10-AWP Bread, experimental bread enriched with 10% (*w*/*w*) of AWP.

**Table 1 foods-12-03743-t001:** Brightness (L*), chroma (C*), and hue angle (h°*) of the CTR and AWP samples. CIELab* values were then converted to RGB. Abbreviation: AWP (avocado waste powder). Results indicate mean values ± S.D. (standard deviation). n.a. = not analyzed. Data within a column followed by different letters are significantly different according to Tukey’s test.

Samples	L*	C*	h°	RGB
Semolina	86.90 ± 1.63 a	32.31 ± 0.72 b	1.35 ± 0.07	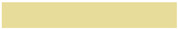
AWP	48.50 ± 1.19 b	50.72 ± 1.90 a	1.52 ± 0.02	
*p* value	<0.0001	<0.0001	0.777	n.a.

**Table 2 foods-12-03743-t002:** Chemical parameters of doughs. Results indicate mean values ± S.D. (standard deviation) of four determinations (carried out in two technical repeats for two independent experiments). Data within a line followed by different letters are significantly different according to Tukey’s test. Abbreviations: TTA, total titratable acidity; CTR, control dough; 5-AWP, experimental dough enriched with 5% (*w*/*w*) of avocado waste powder (AWP); 10-AWP, experimental dough enriched with 10% (*w*/*w*) of AWP; n.a. = not analyzed.

Time	Parameter	Samples				*p*-Value
		Sourdough	CTR	5-AWP	10-AWP	
0 h	pH	3.92 ± 0.04 b	5.50 ± 0.08 a	5.42 ± 0.05 a	5.56 ± 0.08 a	<0.0001
	TTA	12.10 ± 0.07 a	7.58 ± 0.04 b	7.45 ± 0.07 bc	7.33 ± 0.04 c	<0.0001
2 h	pH	n.a.	5.38 ± 0.03	5.34 ± 0.04	5.43 ± 0.13	0.440
	TTA	n.a.	7.63 ± 0.04	7.65 ± 0.07	7.58 ± 0.11	0.565
4 h	pH	n.a.	4.82 ± 0.06 b	4.98 ± 0.01 a	5.07 ± 0.04 a	0.001
	TTA	n.a.	8.08 ± 0.04 a	7.90 ± 0.07 b	7.83 ± 0.04 b	0.003
6 h	pH	n.a.	4.48 ± 0.06 b	4.71 ± 0.02 a	4.70 ± 0.07 a	0.003
	TTA	n.a.	8.50 ± 0.14	8.30 ± 0.07	8.28 ± 0.11	0.092
8 h	pH	n.a.	4.19 ± 0.04 b	4.33 ± 0.02 a	4.37 ± 0.02 a	0.001
	TTA	n.a.	8.93 ± 0.11 a	8.65 ± 0.07 b	8.60 ± 0.07 b	0.007

**Table 3 foods-12-03743-t003:** Microbial loads of doughs. Results indicate mean values ± S.D. (standard deviation) of four plate counts (carried out in two technical repeats for two independent experiments), expressed as Log CFU/g. Data within a line followed by different letters are significantly different according to Tukey’s test. Abbreviations: TMM, total mesophilic microorganisms; LAB, lactic acid bacteria; CTR, control dough; 5-AWP, experimental dough enriched with 5% (*w*/*w*) of avocado waste powder (AWP); 10-AWP, experimental dough enriched with 10% (*w*/*w*) of AWP; n.a. = not analyzed; n.d. = not detected.

Media	Time	Samples				*p*-Value
		Sourdough	CTR	5-AWP	10-AWP	
TMM	0 h	7.76 ± 0.20 a	6.66 ± 0.24 b	6.62 ± 0.13 b	6.60 ± 0.16 b	0.0001
	8 h	n.a.	7.22 ± 0.11	6.76 ± 0.31	7.03 ± 0.14	0.087
Sourdough LAB	0 h	8.64 ± 0.15 a	7.60 ± 0.33 b	7.68 ± 0.28 b	7.71 ± 0.27 b	0.004
	8 h	n.a.	8.94 ± 0.20	8.98 ± 0.19	8.51 ± 0.22	0.055
Yeasts	0 h	7.74 ± 0.25 a	6.52 ± 0.37 b	6.63 ± 0.31 b	6.39 ± 0.27 b	0.002
	8 h	n.a.	7.06 ± 0.16	7.03 ± 0.19	6.78 ± 0.22	0.225
Total coliforms	0 h	<1	<1	<1	<1	n.d.
	8 h	n.a.	<1 b	2.90 ± 0.17 a	3.02 ± 0.28 a	<0.0001
Enterobacteriaceae	0 h	<1	<1	<1	<1	n.d.
	8 h	n.a.	<1 b	3.06 ± 0.32 a	2.80 ± 0.17 a	<0.0001

**Table 4 foods-12-03743-t004:** Quality attributes of bread samples. Results indicate mean values ± S.D. (standard deviation) of four determinations (carried out in two technical repeats for two independent experiments). Data within a line followed by different letters are significantly different according to Tukey’s test. Abbreviations: CTR-Bread, bread produced with control dough; 5-AWP Bread, experimental bread enriched with 5% (*w*/*w*) of avocado waste powder (AWP); 10-AWP Bread, experimental bread enriched with 10% (*w*/*w*) of AWP.

Attributes	Samples			*p*-Value
	CTR-Bread	5-AWP Bread	10-AWP Bread	
Weight loss (%)	11.30 ± 1.37	9.48 ± 0.79	9.94 ± 1.20	0.210
Specific volume (cm^3^/g bread)	3.18 ± 0.11 a	3.09 ± 0.16 a	2.74 ± 0.12 b	0.014
Firmness (N/mm^2^)	0.073 ± 0.007 b	0.102 ± 0.017 a	0.113 ± 0.006 a	0.012
Crust color				
Lightness (L*)	57.07 ± 2.58 a	52.47 ± 1.98 ab	49.87 ± 2.18 b	0.021
Redness (a*)	12.16 ± 1.62 a	5.14 ± 2.70 b	5.39 ± 2.37 b	0.015
Yellowness (b*)	33.58 ± 5.52	36.58 ± 1.61	35.17 ± 0.38	0.574
Crumb color				
Lightness (L*)	71.62 ± 0.87 a	65.37 ± 2.27 b	59.01 ± 0.96 c	<0.001
Redness (a*)	−4.44 ± 0.19 c	−2.80 ± 0.19 b	−1.61 ± 0.17 a	<0.0001
Yellowness (b*)	25.68 ± 0.87	25.20 ± 1.19	24.42 ± 0.93	0.365
Void fraction (%)	34.86 ± 1.75 b	41.14 ± 0.75 a	42.96 ± 2.71 a	0.005
Cell density (n/cm^2^)	58.22 ± 1.54	64.44 ± 10.62	72.59 ± 3.45	0.091
Mean cell area (mm^2^)	0.70 ± 0.05	0.65 ± 0.11	0.59 ± 0.05	0.278

**Table 5 foods-12-03743-t005:** Antioxidant and antiradical activity of semolina and avocado waste powder (AWP). Results indicate mean values ± S.D. Data within a column followed by different letters are significantly different according to Tukey’s test. Abbreviations: AWP (avocado waste powder); TPC (total phenolic content).

Samples	TPC	DPPH_TEAC_	ABTS_TEAC_
	mgGAE/g	mmol TE/100 g	mmol TE/100 g
Semolina	3.676 ± 0.15 b	2.656 ± 0.01 b	2.408 ± 0.04 b
AWP	197.775 ± 0.27 a	38.235 ± 0.09 a	35.175 ± 0.97 a
*p* value	<0.0001	<0.0001	<0.0001

**Table 6 foods-12-03743-t006:** Antioxidant and antiradical activity of fortified and control bread samples (5-AWP Bread, 10-AWP Bread and CTR-Bread). Results indicate mean values ± S.D. Abbreviations: CTR-Bread, bread produced with control dough; 5-AWP Bread, experimental bread enriched with 5% (*w*/*w*) of avocado waste powder (AWP); 10-AWP Bread, experimental bread enriched with 10% (*w*/*w*) of AWP; TPC (total phenolic content).

Samples	TPC	DPPH_TEAC_	ABTS_TEAC_
	mgGAE/g	mmol TE/100 g	mmol TE/100 g
CTR Bread	2.972 ± 0.04	2.311± 0.02	2.102 ± 0.03
5-AWP Bread	23.033 ± 0.38	8.796 ± 0.01	5.985 ± 0.013
10-AWP Bread	23.882 ± 0.09	9.234 ± 0.07	6.656 ± 0.04
*p* value	<0.0001	<0.0001	<0.0001

## Data Availability

The data presented in this study are available on request from the corresponding author.

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
