# Peer review of "A Functional End-Use of Avocado (cv. Hass) Waste through Traditional Semolina Sourdough Bread Production"

_foods, 2023, doi:10.3390/foods12203743_

Round 1
Reviewer 1 Report
The authors aimed to evaluate the effect of partially substituting (5, 10%) wheat (W) semolina with avocado (Persea americana cv. Hass) waste powder (AWP: pulp 50%, seeds/peel 25% each), on the physicochemical (doughs: color, pH, TTA; Breads: Quality parameters), microbiological [total (PC), coliforms (VRBA), yeast (YPD), LAB (mMRS), enterobacteraceae (VRBA) plate counts], sensorial (descriptive, Figure 1) and antioxidant (total phenolics, DPPH/ABTS) profile of sourdough [multiple-strain starter (lines 157-160), 106-107 CFU.g-1, 28 °C for 16 h] and breads [Control (WB) and substituted (5-AWP, 10-APW) breads]. As expected, the partial substitution with AWP produced breads that were slightly more acidic (Table 2) and denser (Table 4), darker, smellier, astringent and with a persistent flavor (Fig 1) and 5-10+ antioxidant profile (Table 6) of sourdoughs with better microbiological quality (Table 3), without an apparent dose-dependent effect. Although the factors (substitution level) and response variables are specific to this study, the findings support those of other similar investigations (e.g., doi: https://tinyurl.com/yfmsru9j, https://tinyurl.com/3zax9xz7). Authors are asked to make some changes in their manuscript to improve its scientific soundness and uniqueness:
General. A) Reading and comprehension of the manuscript will improve if it is reviewed by a native English-speaking colleague or if it is sent to a formal translation agency. B) It is suggested to review once again all the abbreviations used in the manuscript, including their meaning the first time they are mentioned.
Title. Suggestion: “… avocado (cv. Hass) waste…”
Abstract. It is advisable to describe results quantitatively (including p-values) rather than qualitatively, highlighting the most relevant findings from each experimental component and highlighting the dose-response effect of partial substitution (if any).
Introduction. The paragraphs between lines 59-81 basically talk about the thermal treatment of agro-industrial waste and should basically talk about the biotechnological use of waste from the avocado industry, highlighting the fact that these could have technologically convenient characteristics for the baking industry [to reconstruct.
Methods. A) Some sections are too long and detailed, it is advisable to make succinct descriptions (e.g. by referencing methods previously described elsewhere). B) Anticipating that many parameters were evaluated for the same samples, it is advisable to use some indicative statistical tool (e.g. correlation matrix, HCA, PCA, PLS-DA) to more objectively differentiate the benefits and negative effects of partial AWP substitution.
Tables & Figures. A) It is advisable to review once again all tables and figure 1 regarding their statistical descriptors/footnotes and formatting according to the journal´s guidelines. C) Lines corresponding to each sample should be more evident.
Results& Discussion. A greater effort should be made to discuss the results in a more comparative way with previous or similar studies on the subject (partially substituted wheat breads with antioxidant/dietary fiber-rich alternative waste ingredients).
References. A) Double-check the references´ format according to the instructions for the authors, B) Reduce as much as possible the number of old references (currently, 32% of all references ≥10 years old) to say 25%.
Moderate changes are required
Author Response
Dear Reviewer,
corrections and suggestions are marked in yellow in the text.
General.
R. Reading and comprehension of the manuscript will improve if it is reviewed by a native English-speaking colleague or if it is sent to a formal translation agency.
A. The manuscript was revised by an English editing and proofreading service. The English proofreading certificate is attached.
R. It is suggested to review once again all the abbreviations used in the manuscript, including their meaning the first time they are mentioned.
A. Thanks for the suggestion, the text has been revised
R. Title. Suggestion: “… avocado (cv. Hass) waste…”
A. Thanks for the suggestion, the title has been changed
R. Abstract. It is advisable to describe results quantitatively (including p-values) rather than qualitatively, highlighting the most relevant findings from each experimental component and highlighting the dose-response effect of partial substitution (if any).
A. Thanks for the suggestion, the text has been revised
R. Introduction. The paragraphs between lines 59-81 basically talk about the thermal treatment of agro-industrial waste and should basically talk about the biotechnological use of waste from the avocado industry, highlighting the fact that these could have technologically convenient characteristics for the baking industry [to reconstruct.
A. Thanks for the suggestion, the text has been edited
Methods.
R. Some sections are too long and detailed, it is advisable to make succinct descriptions (e.g. by referencing methods previously described elsewhere).
A. The M & M section was summarized as suggested (Sections 2.3, 2.4, 2.7, 2.8, 2.9, 2.11).
R. Anticipating that many parameters were evaluated for the same samples, it is advisable to use some indicative statistical tool (e.g. correlation matrix, HCA, PCA, PLS-DA) to more objectively differentiate the benefits and negative effects of partial AWP substitution.
A. Following your suggestion, a hierarchical cluster analysis was performed. Details on the methodology applied are were reported in paragraph “2.12. Statistical Analysis”, the results shown in Fig. 2 and commented in a new paragraph “3.6. Multivariate Analysis”. The input matrices used for HCA analysis included the relationship between quality attributes, antioxidant and antiradical properties and sensory traits of breads.
Tables & Figures.
R. It is advisable to review once again all tables and figure 1 regarding their statistical descriptors/footnotes and formatting according to the journal´s guidelines.
A. Thank you for the suggestion. Figures and tables have been reformatted according to the guidelines of the journal
R. Lines corresponding to each sample should be more evident.
A. Thanks for the suggestion, the text has been edited
R. Results& Discussion. A greater effort should be made to discuss the results in a more comparative way with previous or similar studies on the subject (partially substituted wheat breads with antioxidant/dietary fiber-rich alternative waste ingredients).
A. Thanks for the suggestion, we will include more information comparing our results with the literature already published
R. References. A) Double-check the references´ format according to the instructions for the authors
A. Thank you for the suggestion. References have been reformatted according to the journal's guidelines.
R. Reduce as much as possible the number of old references (currently, 32% of all references ≥10 years old) to say 25%.
A. Thanks for the advice. The references have been refreshed

Reviewer 2 Report
The aim of the present study was to recycle avocado food waste and by-products by dehydration to produce functional bread. unfortunately, the functionality of the bread was evaluated only based on the total phenolic content and antioxidant activity. In conclusions it is claimed that "The addition of 10 % avocado powder produced doughs with a higher nutritional profile than the control breads". But nutritional profile was not tested in this study (carbohydrates, proteins and lipids). Also it would be valuable to evaluate the changes after the storage.
Other minor comments:
- indicate units of DY (lines 175, 183)
- I think it should be 106 - 107 (lines 175-176)
- indicate units of WL (line 223)
- statistically significant changes must be marked in all tables.
- Table 3 - units is not clear. I'd recommend to write type of microorganism rather then media in the first column.
- Table 5 is not required as all values are given in the text.
- why are the values of CTR in Table 5 and 6 different, as in my opinion they must be identical.
- Latin names must be in "italic".
Author Response
Dear Reviewer,
corrections and suggestions are marked in green in the text.
R. The aim of the present study was to recycle avocado food waste and by-products by dehydration to produce functional bread. unfortunately, the functionality of the bread was evaluated only based on the total phenolic content and antioxidant activity. In conclusions it is claimed that "The addition of 10 % avocado powder produced doughs with a higher nutritional profile than the control breads". But nutritional profile was not tested in this study (carbohydrates, proteins and lipids). Also it would be valuable to evaluate the changes after the storage.
A. Thanks for the advice: "nutritional profile" has been changed to "antioxidant profile"
Other minor comments:
R. indicate units of DY (lines 175, 183)
A. Please consider that this parameter does not have a measure unit as being a ratio between two weights (weight of dough/weight of flour x 100).
R.- I think it should be 106 - 107 (lines 175-176)
A. Modified (Line 228).
R. indicate units of WL (line 223)
A. Added (Line 450).
R. statistically significant changes must be marked in all tables.
A. Following your suggestions, the statistically significant differences among samples were marked in all tables.
R. Table 3 - units is not clear. I'd recommend to write type of microorganism rather then media in the first column.
A. Changed.
R. Table 5 is not required as all values are given in the text.
A. The authors believe it is essential to include a summary table for greater clarity to the reader
R. why are the values of CTR in Table 5 and 6 different, as in my opinion they must be identical.
A. The control values in tables 4 and 5 cannot be the same because they are two different matrices (powders and bread). Furthermore, in the bread samples there is a reduction in the water content which modifies the antioxidant activity values
R. Latin names must be in "italic".
A. Done

Round 2
Reviewer 2 Report
I agree with the corrections within paper. Still minor corrections:
(1) units (Lod CFU/g) must be indicated in the caption of Table 3.
(2) the journal title of ref. [20] must be indicated.
(3) latin names must be written in italic in the reference list
Author Response
Dear Reviewer,
Thank you very much for your suggestions, the requested corrections have been reflected in the text.
- units (Lod CFU/g) must be indicated in the caption of Table 3.
- Done
(2) the journal title of ref. [20] must be indicated.
- Done.
(3) latin names must be written in italic in the reference list
- Done.
